# Oligodendrocytes as A New Therapeutic Target in Schizophrenia: From Histopathological Findings to Neuron-Oligodendrocyte Interaction

**DOI:** 10.3390/cells8121496

**Published:** 2019-11-23

**Authors:** Florian J. Raabe, Lenka Slapakova, Moritz J. Rossner, Ludovico Cantuti-Castelvetri, Mikael Simons, Peter G. Falkai, Andrea Schmitt

**Affiliations:** 1Department of Psychiatry and Psychotherapy, University Hospital, LMU Munich, Nussbaumstrasse 7, 80336 Munich, Germany; Florian.Raabe@med.uni-muenchen.de (F.J.R.); Lenka.Slapakova@med.uni-muenchen.de (L.S.); Peter.Falkai@med.uni-muenchen.de (P.G.F.); 2International Max Planck Research School for Translational Psychiatry (IMPRS-TP), Kraepelinstr, 2-10, 80804 Munich, Germany; 3Molecular and Behavioural Neurobiology, Department of Psychiatry and Psychotherapy, University Hospital, LMU Munich, 80336 Munich, Germany; Moritz.Rossner@med.uni-muenchen.de; 4German Center for Neurodegenerative Diseases (DZNE), Feodor-Lynen Str. 17, 81377 Munich, Germany; Ludovico.Cantuti-Castelvetri@dzne.de (L.C.-C.); Mikael.Simons@dzne.de (M.S.); 5Munich Cluster for Systems Neurology (SyNergy), 81377 Munich, Germany; 6Institute of Neuronal Cell Biology, Technical University Munich, 80805 Munich, Germany; 7Laboratory of Neuroscience (LIM27), Institute of Psychiatry, University of Sao Paulo, 05453-010 São Paulo, Brazil

**Keywords:** schizophrenia, oligodendrocytes, myelin, interneuron, pluripotent stem cells, cognition, treatment

## Abstract

Imaging and postmortem studies have revealed disturbed oligodendroglia-related processes in patients with schizophrenia and provided much evidence for disturbed myelination, irregular gene expression, and altered numbers of oligodendrocytes in the brains of schizophrenia patients. Oligodendrocyte deficits in schizophrenia might be a result of failed maturation and disturbed regeneration and may underlie the cognitive deficits of the disease, which are strongly associated with impaired long-term outcome. Cognition depends on the coordinated activity of neurons and interneurons and intact connectivity. Oligodendrocyte precursors form a synaptic network with parvalbuminergic interneurons, and disturbed crosstalk between these cells may be a cellular basis of pathology in schizophrenia. However, very little is known about the exact axon-glial cellular and molecular processes that may be disturbed in schizophrenia. Until now, investigations were restricted to peripheral tissues, such as blood, correlative imaging studies, genetics, and molecular and histological analyses of postmortem brain samples. The advent of human-induced pluripotent stem cells (hiPSCs) will enable functional analysis in patient-derived living cells and holds great potential for understanding the molecular mechanisms of disturbed oligodendroglial function in schizophrenia. Targeting such mechanisms may contribute to new treatment strategies for previously treatment-resistant cognitive symptoms.

## 1. Introduction

Over 40% of patients with schizophrenia (SZ) have an unfavorable outcome, and only 16% of patients recover with a reduction of symptoms and improvement of social functioning. Cognitive deficits and negative symptoms are the most important predictors for poor social and functional outcome in SZ and major contributors to disability [1,2]. To date, no treatment—including antipsychotics—has shown satisfactory efficacy in improving cognitive deficits and negative symptoms in SZ [3,4], and innovative treatment strategies that target underlying pathological processes are urgently needed.

SZ is regarded as a neurodevelopmental disorder, and risk factors include childhood trauma [5] and birth and obstetric complications [6], all of which may influence brain development. In humans, myelination and white matter development occur at a high rate in the first years of childhood [7], continue at a slightly slower rate during adolescence, and enter another dynamic phase in cortical areas during young adulthood [8], a vulnerable period of brain development that coincides with the average age of onset of SZ [9]. In some cortical areas, myelination contributes to lifelong brain plasticity, an adaptive process to “learning,” and only reaches its maximum level after decades [10].

## 2. Findings of the Relationship of Myelination Deficits, Impaired White Matter, and Cognition from Human Brain Imaging Studies

Diffusion tensor imaging (DTI) examines brain white matter microarchitecture on the basis of free water diffusion properties within a certain three-dimensional area (voxel). Different DTI indices, or their combinations, are associated (to different extents) with the degree of fiber tract organization and myelination and with axonal integrity. The most important DTI indices are fractional anisotropy (FA), mean diffusivity (MD), axial diffusivity (AD), and radial diffusivity (RD) [11]. A pioneer study performed DTI imaging and CLARITY immunolabeling of whole-brain myelin basic protein (MBP), which is essential for myelination and represents 30% of total myelin brain protein [12], in the same mouse brain, and revealed that DTI-derived FA significantly correlated with MBP expression, whereas MD, AD, and RD did not [13]. Previous meta-analyses of voxel-based DTI studies in SZ and first-episode SZ found reduced FA, in particular in fronto-temporal-limbic pathways [14,15,16].

Recently, the ENIGMA Schizophrenia DTI Working Group performed the largest international multicenter study to date, in 4322 individuals, which revealed broad white matter microstructural differences in SZ [17]. Widespread FA reductions were significant in 19 of 25 regions and, remarkably, were driven more by peripheral white matter disturbances than by disturbances in specific core regions of interest. The group also found a widespread increase in MD and RD and consequently suggested that lower FA was most likely driven by aberrant myelination in most regions [17]. The group concluded that lower FA in SZ is not due to the potential impact of antipsychotic treatment on white matter because it did not find significant associations between FA and antipsychotic treatment [17]. This conclusion is supported by a meta-analysis in minimally treated first-episode SZ patients that also showed a reduction in FA in the fronto-limbic circuitry [15].

Disturbances of white matter integrity within the fornix, which contains the white matter tracts of the hippocampus, correlate with impairments in episodic memory in SZ [18,19]. Oligodendrocyte dysfunction leads to disturbances in myelination and consequently to deficient propagation of nerve impulses and impaired cognition [20]. Moreover, in SZ patients and controls, oligodendrocyte-related gene variants, such as myelin-associated glycoprotein (MAG) and cyclic nucleotide phosphodiesterase (CNP), are related to cognitive performance and this relationship is mediated by white matter tract integrity [21]. Interestingly, a single nucleotide polymorphism of the oligodendrocyte transcription factor Olig2, which is necessary for maturation of oligodendrocyte progenitor cells, has also been associated with impaired cognition, mediated by reduced white matter FA in healthy controls [21]. FA and cognition are also reduced in individuals at ultra-high risk for psychosis, and lower FA, accompanied by higher RD, was linked to demyelination [22]. Moreover, widespread higher FA was associated with improved cognitive performance in people at ultra-high risk for psychosis, but not in healthy controls [22].

## 3. Histopathological Studies of Oligodendrocytes in Schizophrenia (SZ)

A reduction of perineuronal oligodendrocytes in the gray matter of the prefrontal cortex has been reported in SZ [23]. Stereological analyses have found a reduced number of oligodendrocytes in the dorsolateral prefrontal cortex (DLPFC) [24] but not in the anterior cingulate cortex [25]. In design-based stereological postmortem studies of Nissl (cresyl violet) and myelin (luxol fast-blue) stained sections, our group showed a decreased oligodendrocyte number in the left CA4 region of the anterior and posterior hippocampus in SZ [26,27]. The stereologically estimated loss of oligodendrocytes in this region was associated with cognitive deficits [28]. A study that aimed to validate the loss of oligodendrocytes by using immunohistochemical markers found a trend for decreased oligodendrocyte transcription factor Olig1 immuno-positive oligodendrocyte density in the left CA4, but no reduction of the transcription factor Olig2 [28]. Olig1 antibodies stain precursor forms and mature oligodendrocyte populations, and both Olig1 and Olig2 are needed for progenitor development and repair of myelin [29]. Moreover, the finding by Schmitt et al. (2009) [26] and Falkai et al. (2016) [27] of decreased oligodendrocyte number and Falkai et al. (2016) [28] of association with cognitive deficits led to the hypothesis that the decreased number of oligodendrocytes is related to a failure of maturation and indicates a disturbed regenerative recovery process in the CA4/dentate region [30]. Interestingly, the loss of oligodendrocytes is confined to the CA4 region, a region that is now regarded as the polymorph layer of the dentate gyrus. This region connects the dentate gyrus, where neurogenesis can be observed, with the CA3 region [31]. We found evidence for disturbed neurogenesis in SZ in that the volume and number of granule neurons in the left dentate gyrus were reduced [28]. These findings replicated those of former studies [32] that described such thinning and were interpreted as a sign for disturbed neurodevelopment in SZ. Furthermore, the CA4/dentate gyrus region is the neuroanatomical basis for the cognitive domain “pattern separation” and other neurocognitive functions such as declarative memory, which have been shown to be disturbed in SZ [33].

When interpreting histopathology studies in SZ, one must also consider their limitations. Design-based stereological studies are superior to cell density studies because the two-dimensional assessment of Olig1 or Olig2 immunostained cells may be confounded by volume differences that are due to tissue shrinkage associated with formalin fixation or staining procedures, cutting of cells during sectioning, non-random orientation, and irregular cell shape and size [34]. Moreover, long-term treatment with antipsychotics may confound results. Using design-based stereology in histologically stained serial brain sections, we performed a count of the different cell types based on morphological criteria (neurons, astrocytes, oligodendrocytes) that come into focus within unbiased virtual counting spaces distributed in a systematic-random fashion throughout the different regions of the hippocampus. Estimated cell numbers were calculated from the numbers of counted cells and the sampling probability according to Schmitz and Hof (2005) [35]. Our group showed that the dose of antipsychotics in chlorpromazine equivalents had no influence on oligodendrocyte numbers [26,27].

## 4. Evidence of Oligodendrocyte Deficits from Molecular Studies

In SZ, genome-wide microarray studies have shown that expression of myelin- and oligodendrocyte-related genes is profoundly affected in the prefrontal, temporal, and occipital cortex, hippocampus, and basal ganglia [36].

In a microarray study, in the temporal cortex, our group showed decreased mRNA expression of contactin-associated protein, which mediates contact between oligodendrocytes and the synapse, thus indicating dysfunctional oligodendrocyte-neuronal interactions in SZ (Schmitt et al. 2012). In a series of proteomic studies in frozen postmortem tissue, we showed that in SZ myelination-related proteins, such as MBP and myelin oligodendrocyte glycoprotein (MOG), are downregulated in the DLPFC, anterior temporal lobe, and corpus callosum (e.g., [37,38]). In an immunohistochemistry study, we detected a decreased intensity of myelin-related MBP staining in the entorhinal cortex of SZ patients and found a correlation between decreased myelination and disorganization of pre-alpha cells [39]. Single-cell transcriptome analysis of gene expression in different cell populations [40], such as oligodendrocytes in the hippocampal region and prefrontal cortex, has not yet been performed in postmortem brains from patients with SZ.

## 5. The “Defective Maturation” Hypothesis of SZ

To date, it is unclear whether a loss of oligodendrocyte progenitors or of mature oligodendrocytes, and therefore a failure in differentiation or apoptosis, contributes to the reduced number of oligodendrocytes in patients with SZ. The cause of reduced oligodendrocyte numbers may be important for the development of future treatment strategies targeting deficits in oligodendrocyte-related pathological processes. For example, one potential treatment may be to improve the differentiation of oligodendrocyte progenitor cells to myelinating oligodendrocytes, thereby promoting remyelination and possibly contributing to improvement of treatment-resistant cognitive and negative symptoms.

Animal models have shown that oligodendrocyte progenitor cell proliferation and differentiation is required for remyelination [41]. However, in multiple sclerosis, remyelination is often incomplete. Besides a loss of mature oligodendrocytes, reductions in oligodendrocyte progenitor cells have been reported [42], as well as increased death of these progenitor cells and reduced process extension under stress conditions [43]. It has been hypothesized that oligodendrocyte progenitor cells, which are capable of myelination, are reduced in brain regions of SZ patients, resulting in decreased plasticity and remyelination capacity. Progenitor cells can be labeled by using antibodies that bind to oligodendrocyte proteins, which are expressed during specific stages of oligodendrocyte development [41]. However, a first cell density study of the prefrontal cortex in SZ detected no loss of early NG2-immunopositive oligodendrocyte progenitor cells [44]. This study did detect a loss of oligodendrocytes positive for Olig2, a transcription factor expressed in oligodendrocyte progenitors at later stages and in mature oligodendrocytes [44], but Olig2 is not suitable for identifying progenitor cells specifically. Additional labeling with neurite outgrowth inhibitor (Nogo)-A, which reliably identifies mature oligodendrocytes, has been shown to be a way to identify specific stages of oligodendrocytes in human brain regions from patients with multiple sclerosis [45]. Nogo is known to regulate cellular processes and has three isoforms, Nogo-A, -B, and -C. Specifically, Nogo-A is highly expressed in oligodendrocytes. Mature oligodendrocytes derived from surgery tissue specimens from adult patients express both Nogo-A and Olig2. Double immunohistochemistry with anti-Nogo-A, a marker that reliably identifies mature oligodendrocytes in human CNS tissue [45], revealed that almost all of the weakly positive Olig2 cells were also Nogo-A positive and were identified as mature oligodendrocytes. In contrast, Olig2-strong cells were negative for Nogo-A. Therefore, double-staining immunohistochemistry allows oligodendrocyte progenitors to be reliably identified and studies identified oligodendrocytes with weak Olig2 and strong NogoA staining as mature oligodendrocytes, but those with strong Olig2 and negative NogoA staining as progenitors [43,46]. Other immunohistochemical markers, such as PDFαR and NG2, have also been used to identify oligodendrocyte progenitor cells [41]. In SZ, however, stereological studies investigating the number and apoptosis of mature oligodendrocytes or progenitors are still lacking.

## 6. The Intercellular Interactions of Oligodendrocytes with Microglia and Neurons

A meta-analysis reported mild inflammation of the brain in SZ with activation of microglia [47], which may contribute to the oligodendrocyte deficit [48]. Ultrastructural analysis revealed activated microglia near dystrophic and apoptotic oligodendrocytes and demyelinating and dysmyelinating axons [49,50]. Oligodendrocytes have glutamatergic n-methyl-D-aspartate (NMDA) receptors, and our group showed that NMDA receptor hypofunction after MK-801 treatment of human cell cultures causes oligodendrocyte dysfunction by inducing deficits in glycolysis [51]. MK-801 is a potent NMDA receptor antagonist, and treatment with this class of antagonists represents the most reliable pharmacological model of the cognitive, positive, and negative symptoms of SZ [52,53]. Therefore, NMDAR antibodies, as part of an inflammatory process, may influence oligodendrocyte pathology.

In SZ, a dysfunction of γ-amino-butyric acid (GABA)ergic interneurons has been proposed to play a role in the pathophysiology of cognitive deficits [54]. More specifically, mRNA and protein levels of parvalbumin-positive interneurons were shown to be affected in SZ, while cell number and density were not consistently reduced [55]. However, a dysfunction of inhibitory interneurons may contribute to a hypofunction of the NMDA receptor and a glutamatergic imbalance, leading to cognitive deficits and negative and positive symptoms [52]. Recently, it has become evident that, besides the well-known myelination of glutamatergic projection neurons, a large fraction of myelin ensheathes axons of cortical inhibitory neurons, especially of parvalbumin-positive basket cells [56]. These findings have relevance for oligodendrocyte pathology in SZ [55] because synaptic signaling between interneurons and oligodendrocyte precursor cells is known to influence the differentiation of oligodendrocyte progenitors in the hippocampus [57].

The dysfunction of parvalbuminergic interneurons may be a result of impaired myelin plasticity. Fast-spiking parvalbuminergic interneurons are essential in generating cortical oscillations in the gamma range (30–120 Hz), mediated by synchronized inhibition of pyramidal neurons [58,59]. Through rhythmic perisomatic inhibition of pyramidal neurons, synchronous ensembles of parvalbuminergic interneurons evoke high-frequency gamma oscillations in the cortex and hippocampus [60,61]. These gamma oscillations can be determined by electroencephalographic (EEG) recordings [62]. In SZ, dysfunctional gamma oscillations are the basis of deficits in cognitive functions, such as attention and working memory [63,64,65]. Impaired maturation of interneuron-related gamma oscillations may be a fundamental link between the cognitive and memory deficits associated with early life stress and the etiologies of SZ, which are based on aberrant neurodevelopment [66].

The relationship between oligodendrocytes and interneuron pathology in SZ is unknown [67]. A large fraction of myelinating oligodendrocytes ensheath fast-spiking parvalbuminergic interneurons. The fast-spiking parvalbumin-positive inhibitory interneurons of the basket cell class, which have very high tonic activity, may require myelin to support their high-energy demands [68], and it is presumed that myelin regulates extracellular potassium buffering [69,70]. Glycolytic oligodendrocytes support the energy demands of axonal intermediate metabolism by delivering lactate to the encapsulated axonal compartment, so that neuronal mitochondria can generate ATP [71,72]. Moreover, optogenetic activation of parvalbumin-positive interneurons in the mouse primary visual cortex (V1) sharpens neuronal feature selectivity and improves perceptual discrimination, and therefore, parvalbuminergic activation has functional and behavioral impact [73]. Lactate needs to be delivered because myelin prevents axons from having rapid access to extracellular metabolites. This concept of metabolic coupling of myelin and axons is an important new development in neuroscience [74]. Besides myelination and metabolic support, electrically coupled perisomatic oligodendrocytes buffer K+ currents and influence high-frequency neuronal excitability, e.g., of excitatory pyramidal [69] and hippocampal inhibitory interneurons [75].

## 7. The Role of Environmental Risk Factors in Oligodendrocyte Differentiation

Myelination of the brain has been shown to depend on experiences, and neurodevelopmental stress-related disturbances in social experience-dependent myelination have been proposed to play a role in SZ [76]. The mouse model of social isolation immediately after weaning (postnatal day 21–50) presents with a deficit in oligodendrocyte morphology, reduced myelin thickness, decreased MBP and MAG expression, a deficit in SZ-related behavior (PPI, working memory), and decreased social exploration [77,78]. Importantly, in contrast to the effects of adult social isolation, this early induced phenotype could not be rescued by later social re-integration [78]. In adult mice exposed to social isolation, clemastine, a muscarinic receptor antagonist, enhanced oligodendrocyte differentiation and myelination and improved social avoidance behavior [79].

Epidemiological studies have proven that exposure to early stress in the form of abuse and neglect in childhood increases the risk for later development of SZ [5,80,81]. Childhood abuse and neglect are known to have a negative influence on cognition in patients with SZ [5]. However, to date, no specific treatment exists for SZ-related cognitive deficits, negative symptoms, and underlying myelination and oligodendrocyte deficits. In this context, drug repurposing is a promising way to address new treatment targets aimed at improving outcome in SZ. Miconazole (an antifungal agent) and clobetasol (a corticosteroid) are known to improve remyelination and maturation of oligodendrocytes, and the latter is also an immunosuppressant. In the lysolecithin lesion model, a multiple sclerosis mouse model, both substances enhanced remyelination and increased the number of new oligodendrocytes. Moreover, these drugs enhanced differentiation and maturation of oligodendrocytes in mouse pluripotent epiblast stem cell-derived oligodendrocyte progenitor cells [82].

## 8. The Impact of Genetic Schizophrenia Risk on the Oligodendroglial Linage

Genome-wide association studies (GWAS) and exome sequencing approaches have provided solid evidence of common and rare genetic variations in complex psychiatric disorders such as SZ. So far, around 150 genetic risk single nucleotide polymorphisms (SNPs) have been unequivocally identified [83], and more loci will be revealed by the most recent GWAS studies with increased sample sizes. GWAS have validated the polygenic architecture of SZ, which was postulated decades before [84]. Further analysis identified several risk SNPs associated with genes of known regulatory function in neurons and also SNPs associated with genes relevant for glial cells, oligodendrocyte progenitor cells, and mature oligodendrocytes [85,86,87]. Remarkably, the expert-curated glia-oligodendrocyte pathway (comprising 52 genes) is associated more strongly with the genetic risk for SZ than with that for bipolar disorder [86]. In a study of uncurated but computed cell type-specific gene expression based on mice scRNA-seq and human snRNAseq data, SZ risk genes identified in GWAS were most significantly associated with a dedicated set of mature neuronal cell types (medium spiny neurons, cortical and hippocampal glutamatergic projection neurons, and cortical GABAergic interneurons) than with other neuronal or glial cell types [87]. However, based on only human cell-type specific gene expression profiles, oligodendrocyte progenitor cells and oligodendrocytes also showed enrichment in genes associated with SZ. In this study, the cell-type association of astrocytes or microglia was much lower [87]. Interestingly, increasing evidence indicates that aerobic exercise increases hippocampal volume and improves cognition in SZ patients [88,89,90]. Previous studies showed that the effects of exercise on the hippocampus might be connected to the polygenic burden of SZ risk variants [89]. The modulatory role of cell type-specific SZ polygenic risk scores (PRS) on exercise-induced volume changes in the CA1, CA2/3, and CA4/dentate gyrus subfields was recently assessed. These analyses showed that the polygenic burden associated with oligodendrocyte precursor cells and radial glia significantly influenced the volume changes between baseline and three months in the CA4/dentate gyrus subfield in SZ patients performing endurance training. A higher oligodendrocyte precursor cell- or RG-associated genetic risk burden was associated with a less pronounced volume increase or even a decrease in CA4/dentate gyrus during the exercise intervention. Therefore, it was hypothesized that SZ cell type-specific polygenic risk modulates the aerobic exercise-induced neuroplastic processes in CA4/dentate gyrus of the hippocampus [91].

## 9. Patient-Derived Neurobiological Test Systems Indicate Oligodendroglial Contribution to SZ

Until recently, most insights into SZ have been generated from postmortem tissue samples and imaging, genetic, pharmacological, and animal studies. Cellular reprogramming methods to generate induced pluripotent stem cells (iPSC) now provide a new opportunity to model the complex polygenetic conditions of SZ by generating patient-derived human iPSC (hiPSC)-based neurobiological test systems [92,93]. The pioneer work of Brennand et al. (2011) first characterized hiPSC-derived neurons from SZ patients and revealed decreased neuronal connectivity, decreased neurites, and decreased levels of post-synaptic protein PSD95 [94]. Subsequent studies focused on specific neuronal subtypes, such as pyramidal cortical interneurons and dentate gyrus granule neurons, and a series of studies revealed cell-autonomous neuronal disturbances in SZ [92,95]. Although pioneer studies confirmed postmortem findings and revealed additional aspects of the molecular mechanisms of SZ, hiPSC-based disease modeling has several limitations. Economical and technical limitations include high costs, biological intra- and inter-individual variability, robustness of applied protocols, affordability, and scalability. Most studies included fewer than five individuals per group, and only a few included more than 10 individuals per group. However, the field of hiPSC is rapidly evolving and is addressing the above-mentioned challenges. Nevertheless, several conceptual limitations will remain, at least in the medium term. Examples of such conceptual limitations are as follows: (1) hiPSC-based systems cannot fully mimic the human gene x real world environment interactions that are part of the etiology of SZ [96], although aspects of known environmental risk factors (e.g., infection, stress, inflammation) can be modeled [95]; (2) hiPSC-based models are more powerful models of genetic risk for SZ than of SZ as a disease entity; (3) hiPSC models do not mimic network macro connectivity, which is assumed to be disturbed in SZ [97]; and, (4) long-lasting processes, such as aging and maturation over many years, are disturbed in SZ [98] but are difficult to mimic in vitro.

In contrast to investigations on hiPSC-derived neurons, only very few studies have investigated the impact of oligodendroglial cells in SZ-related hiPSC models. Expression of the SZ risk gene FEZ1 is regulated by SZ-relevant pathways, and knockdown of FEZ1 in murine and human iPSC-derived oligodendroglial cells was found to disturb oligodendrocyte development [99]. A family-based approach used hiPSC oligodendrocyte progenitor cells to investigate the contribution to SZ of two rare missense mutations in CSPG4, which codes for NG2, a prominent marker for proliferating oligodendrocyte progenitor cells [100]. The study found that hiPSC oligodendrocyte progenitor cells with one of the CSPG4 mutations showed dysregulated posttranslational processing, subcellular localization of mutant NG2, and impaired oligodendrocyte progenitor cell survival, with reduced differentiation to mature oligodendrocytes. Carrier-derived hiPSC neurons were not pathological, underlining the oligodendroglial cell-autonomous effect of the CSPG4 mutations. Remarkably, DTI-detectable impairments of white matter integrity were found in affected mutation carriers but not in their unaffected siblings or the general population [100]. In a pioneer study by Windrem and colleagues [101] in hiPSC from patients with childhood-onset SZ, glial precursor cells, which could mature into both oligodendroglial and astroglial lineage cells, showed altered transcriptomic signatures and impaired astroglial maturation and hypomyelination. Moreover, immune-deficient mice that received human precursor cells from SZ patients showed psychosis-related behaviors and cognitive impairments compared with control mice that received cells from healthy individuals [101]. Another study revealed reduced differentiation of hiPSC-derived marker O4 of the oligodendrocyte lineage (O4-positive cells) late oligodendrocyte progenitor cells and oligodendrocytes in SZ patient-derived hiPSC lines compared with control lines. Moreover, white matter myelin content correlated with the number of O4-positive cells [102]. The above studies underline the cell-autonomous contribution of the oligodendroglial lineage to SZ. However, they have several limitations. Family-based studies investigated single, rare SZ variants with large effects [99,100], but the genetic reality of most SZ patients is a polygenic accumulation of common variants with low individual effect sizes [83]. Windrem et al. studied glial progenitor cells (GPCs) in a very limited number of individuals with childhood-onset SZ (a rare disorder) with a very time-consuming experimental protocol (>200 days to generate GPCs) [101], which limits subsequent functional analysis or rescue experiments. McPhie et al. found evidence for impaired development of oligodendrocytes in SZ, but their analysis was limited to immunocytochemistry and did not dissect possible underlying mechanisms [102]. All these pioneering studies used different approaches that needed 65 to more than 200 days. Therefore, studies are needed that pave the way for modeling diseases within a shorter time and thus enable the cell-type specific dissection of disturbed pathways, gene regulation, and molecular mechanisms in a more systematic and potentially scalable manner.

Technically, and similar to the case with neurons, two different strategies are available to generate hiPSC-derived oligodendrocyte progenitor cells/oligodendrocytes (for details, we refer the reader to detailed reviews [103,104]). The first and older strategy is to mimic the embryological and “natural” development of oligodendrocyte progenitor cells/oligodendrocytes by in vitro patterning with chemical stimulation. The advantage of this method is that researchers can investigate the developmental aspect of a disease. The disadvantages are the time (it takes 55 to more than 200 days to generate O4+ late-stage oligodendrocyte progenitor cells) and costs of generating oligodendrocyte progenitor cells/oligodendrocytes. Recent developments have tried to accelerate extracellular lineage pattering by adding ectopic expression of cell-type determining transcription factors [105,106]. This approach allows hiPSCs to be differentiated to MBP+ oligodendrocytes within 22 days [106]. An additional advantage is the reduced cellular heterogeneity. Probably the most important disadvantages of directed differentiation approaches are their limitations in studying the early developmental aspects of SZ [93]. Oligodendrocyte progenitor cells and oligodendrocytes are heterogeneous across brain regions and vary with age [107], so investigations are needed that address this diversity.

## 10. The Road to New Therapies

Imaging, postmortem, and pioneer hiPSC studies have provided evidence for cell-autonomous deficits of the oligodendroglial lineage in SZ (Table 1). Despite the tremendous progress in two- and three-dimensional hiPSC-derived myelinating neurobiological test systems, these systems are always limited by their construct validity in brain disorders, where circuit levels contribute to behavioral and cognitive deficits. Nevertheless, patient-specific cellular systems enable the study of disease-associated endophenotypes, such as axonal support or multiple aspects of myelination, and expand the experimental repertoire in psychiatric research [93].

Besides technical and conceptual limitations of hiPSC-based disease modeling of a complex disease such as SZ, a major challenge in generating useful patient-derived neurobiological test systems is meaningful patient stratification [93]. Future translational studies need to investigate the characteristics of such stratification. A stringent, at best hypothesis-driven pre-selection of relevant patient subgroups might allow corresponding molecular mechanisms to be identified in SZ. In addition to human and animal in vivo studies, hiPSC technology might be a key method to identify diseases-relevant cellular and molecular profiles and to perform subsequent genetic and pharmacological rescue experiments (Figure 1). Despite important limitations, hiPSC-based disease modeling represents a new and potentially powerful option to study cellular phenotypes in SZ. hiPSC technology allows researchers to use personalized strategies to address old questions and might help identify different molecular pathways as potential targets for new treatment strategies.

## Figures and Tables

**Figure 1 cells-08-01496-f001:**
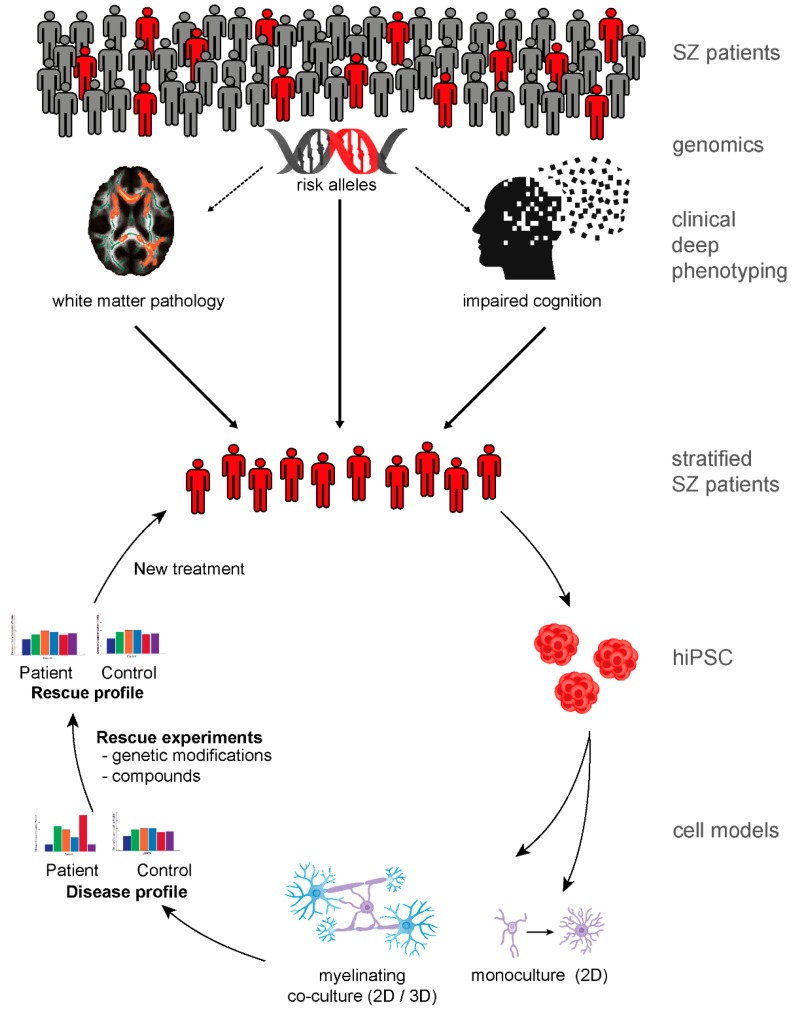
Principals of patient stratification for subsequent human-induced pluripotent stem cell (hiPSC)-based cellular disease modeling and new treatment strategies. Stratification of schizophrenia (SZ) patients could be based on genetics or endophenotypes or a combination of the two. Recent evidence suggests that patients with oligodendrocyte dysfunction and white matter pathology have cognitive impairments. Red human icons illustrate patients who are risk gene carriers with the shared endophenotypes of disturbed white matter pathology and impaired cognition. Meaningful patient stratification based on genomics and clinical deep phenotyping enables subsequent investigations of underlining cellular and molecular mechanisms. hiPSC technology enables the generation of a toolbox of patient-derived cell models. Monocultures of glial cells and myelinating co-culture systems could simulate disease-relevant endophenotype profiles of SZ in vitro. Moreover, hiPSC-derived models can be used for genetic and pharmacological rescue experiments and pave the way for new treatment options. Aspects or parts of the illustrations have been published previously [93,111].

**Table 1 cells-08-01496-t001:** Summary of disturbed oligodendrocyte function in schizophrenia. CA4: cornu ammonis 4; DLPFC: dorsolateral prefrontal cortex; hiPSC: human induced pluripotent stem cells; iPSC: induced pluripotent stem cells; MAG: myelin-associated glycoprotein; MBP: myelin basic protein; MOG: myelin oligodendrocyte glycoprotein; SZ: schizophrenia.

In vivo brain imaging studies	Decreased fractional anisotropy as a sign of impaired white matter tract integrity [14,17] Deficits in connectivity in relevant neuronal networks [108] Single nucleotide polymorphisms in the *MAG* and *Olig2* genes are related to white matter tract integrity and cognitive performance [21]
Histopathology (postmortem)	Decreased oligodendrocyte number in DLPFC and CA4 of the hippocampus [24,26,27] Decreased MBP immunohistochemical staining intensity [39] Reduced density of perineuronal oligodendrocytes [23]
Transcriptomic studies	Decreased expression of myelin- and oligodendrocyte-related genes, such as *MAG* and *MBP*, in several relevant brain regions [109,110]
Proteomic studies	Decreased expression of myelin- and oligodendrocyte-related proteins, such as MOG and MBP, in several relevant gray and white matter brain regions [37,38]
hiPSC studies	Impaired oligodendrocyte maturation and hypomyelinization after neonatal implantation into mice of iPSC-derived oligodendrocyte progenitor cells from SZ patients [101] Reduced differentiation of O4-positive late oligodendrocyte precursor cells and oligodendrocytes from SZ hiPSC lines compared with control hiPSC lines. Correlation between white matter myelin content and number of O4-positive cells [102]

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
