# Peer review of "Oligodendrocytes as A New Therapeutic Target in Schizophrenia: From Histopathological Findings to Neuron-Oligodendrocyte Interaction"

_cells, 2019, doi:10.3390/cells8121496_

Round 1

Reviewer 1 Report

This review focuses on the potential involvement of oligodendrocytes in the pathobiology of schizophrenia.

The review is well thought, well constructed and the authors made a great effort in term of documentation and references. The reading of the text is attractive and pleasant.

the main concern is that the authors should discussed better the limitations and the future challenges required for using human pluripotent stem cells for modeling SZ.

Minor comments :

The last chapter of the review concerns the potential of human pluripotent stem cells derived from patients to better decipher the pathobiology involved in SZ. The authors should however raised the limitations of the studies that have been published on the subject. As an example, the authors should discuss the limitations about the use of human pluripotent stem cells for modeling complex diseases. The authors should also discuss about the protocols that actually exist for differentiating human pluripotent stem cells into ologidendrocytes. For my knowledge, there are long and not efficient which might be another limitations for modeling the impact of SZ in this cell type. Another point that should be discuss is the obtention of regional oligodendrocytes. As the authors mentioned, the affection of oligodendrocytes seems to be region specific which implies the capacity to produce from human pluripotent stem cells regionalized ologodendrocytes. Some turns-of-phrase should be improved. As an example, the phrase on DTI indices (line 63) is long and should be divided into two separate phrases to a better clarity.

Author Response

We thank the reviewers for the time, thoughts, and effort they put into reviewing our manuscript and for their suggestions on how we can improve it. We have revised the text on the basis of these suggestions. In addition to addressing the reviewer’s suggestions, we have added longer paragraphs on the limitations of hiPSC-based SZ modelling and a short technical section to explain the principals of hiPSC-derived oligodendrocyte precursor cell/oligodendrocyte differentiation. Moreover, the revised version has been edited. Please consider that reference numbers have changed because we have added a total of 12 references.

Response to Reviewer 1

This review focuses on the potential involvement of oligodendrocytes in the pathobiology of schizophrenia.

The review is well thought, well constructed and the authors made a great effort in term of documentation and references. The reading of the text is attractive and pleasant.

the main concern is that the authors should discussed better the limitations and the future challenges required for using human pluripotent stem cells for modeling SZ.

We thank the reviewer and agree with this remark. In response, we have added text to the section “Patient-derived neurobiological test systems…” to highlight technical and conceptual limitations of hiPSC-based disease modeling in general, as well as future technical challenges. We discuss the limitations of published studies that investigated oligodendrocyte dysfunction in SZ.

The new paragraph (page 7, line 312-325) reads as follows: “Although pioneer studies confirmed postmortem findings and revealed additional aspects of the molecular mechanisms of SZ, hiPSC-based disease modeling has several limitations. Economical and technical limitations include high costs, biological intra- and inter-individual variability, robustness of applied protocols, affordability, and scalability. Most studies included fewer than 5 individuals per group, and only a few included more than 10 individuals per group. However, the field of hiPSC is rapidly evolving and is addressing the above-mentioned challenges. Nevertheless, several conceptual limitations will remain, at least in the medium term. Examples of such conceptual limitations are as follows: 1) hiPSC-based systems cannot fully mimic the human gene x real world environment interactions that are part of the etiology of SZ [97], although aspects of known environmental risk factors (e.g. infection, stress, inflammation) can be modeled [96]; 2) hiPSC-based models are more powerful models of genetic risk for SZ than of SZ as a disease entity; 3) hiPSC models do not mimic network macro connectivity, which is assumed to be disturbed in SZ [98]; and, 4) long-lasting processes, such as aging and maturation over many years, are disturbed in SZ [99] but are difficult to mimic in vitro.”

Minor comments:

The last chapter of the review concerns the potential of human pluripotent stem cells derived from patients to better decipher the pathobiology involved in SZ. The authors should however raised the limitations of the studies that have been published on the subject. As an example, the authors should discuss the limitations about the use of human pluripotent stem cells for modeling complex diseases.

We thank Reviewer 1 for encouraging us to highlight the important limitations of hiPSC-based SZ modeling.

First, we added a long paragraph on this important issue (Section “Patient-derived neurobiological test systems…”; please see response above).

Second, we added a paragraph in which we discuss limitations of the published studies that investigate the oligoglial contribution in hiPSC-based systems. Page 8, line 348-360 reads as follows: “However, they have several limitations. Family-based studies investigated single, rare SZ variants with large effects [100,101], but the genetic reality of most SZ patients is a polygenic accumulation of common variants with low single effect size [85]. Windrem et al. studied glial progenitor cells (GPCs) in a very limited number of individuals with childhood-onset SZ, (a rare disorder) with a very time-consuming experimental protocol (> 200 days to generate GPCs) [102], which limits subsequent functional analysis or rescue experiments. McPhie et al. found evidence for impaired development of oligodendrocytes in SZ, but their analysis was limited to immunocytochemistry and did not dissect possible underlying mechanisms [103]. All these pioneering studies used protracted differentiation approaches that needed 65 to more than 200 days. Therefore, studies are needed that pave the way for modeling diseases within a shorter time and thus enable the cell-type specific dissection of disturbed pathways, gene regulation, and molecular mechanisms in a more systematic and potentially scalable manner.”

The authors should also discuss about the protocols that actually exist for differentiating human pluripotent stem cells into ologidendrocytes. For my knowledge, there are long and not efficient which might be another limitations for modeling the impact of SZ in this cell type.

We agree with this comment and have included a paragraph that briefly highlights current strategies to generate hiPSC-derived OPC/OL. For technical details, we refer the reader to reviews that focus on this technical issue. Page 8, line 362-375 reads: “Technically, and similar to the case with neurons, two different strategies are available to generate hiPSC-derived oligodendrocyte progenitor cells/oligodendrocytes (for details, we refer the reader to detailed reviews [104,105]). The first and older strategy is to mimic the embryological and “natural” development of oligodendrocyte progenitor cells/oligodendrocytes by in vitro patterning with chemical stimulation. The advantage of this method is that researchers can investigate the developmental aspect of a disease. The disadvantages are the time- (it takes 55 to more than 200 days to generate O4+ late-stage oligodendrocyte progenitor cells) and costs of generating oligodendrocyte progenitor cells/oligodendrocytes. Recent developments have tried to accelerate extracellular lineage pattering by adding ectopic expression of cell-type determining transcription factors [106,107]. This approach allows hiPSCs to be differentiated to MBP+ oligodendrocytes within 22 days [107]. An additional advantage is the reduced cellular heterogeneity. Probably the most important disadvantages of directed differentiation approaches are their limitations in studying the early developmental aspects of SZ [94]. Oligodendrocyte progenitor cells and oligodendrocytes are heterogeneous across brain regions and vary with age [108], so investigations are needed that address this diversity.”

Another point that should be discuss is the obtention of regional oligodendrocytes. As the authors mentioned, the affection of oligodendrocytes seems to be region specific which implies the capacity to produce from human pluripotent stem cells regionalized ologodendrocytes.

We thank the reviewer for this comment and now highlight this important aspect and unmet need in the paragraph on technical issues (see above).

Some turns-of-phrase should be improved. As an example, the phrase on DTI indices (line 63) is long and should be divided into two separate phrases to a better clarity.

We have clarified certain phrases in our manuscript and performed a second language editing.

Reviewer 2 Report

There are many hypothetic ideas about using hiPSCs.

The isdea of the survey is great, but if the views about interractions between cells in the brain regions discussed are interesting, especially with reafgrds to neurotransmitters and other interreralationships, the imagery is poor to make such hypothesis about hiPSCs review.

Readxing and reviewing some references cited, such as 24 and 25, micrographs there are deceiving because the preservation of tissue oligodendrocytes was poor and accepted for publication! The images seemed to indicate that the oligodendrocytes of normal brain patients are in fact quite similar to those of schizophrenic patients!  Moreover, would those published micrographs really represented oligodendrocytes?

Additionally, to claim apoptosis...which is programmed cell death...how to justify this term and functions, especially if demyelination showed is not centrally located and without nucleus with apoptotic views? More, what would be apoptosis there, why not necrosis? how about necroptosis?

The cells illustrated are not typical, certainly not for normal patients because oligodendrocytes have electron dense cytoplasms and both are clear and damaged (i.e. control vs schizophrenic ones).  Infact if the imagery and statistics can satisfy the peripheral or nerve myelinated micrographs, the support for cell bodies is none.

Similarly, one of the reference, i.e. 32, showed in morphometry histograms, that the oligodendrocytes not dissimilar to control ones due to scattered bar of errors in measurements.

I would suggest a Table 1, basec on imagery like the entire manuscript is questionable and should await further not so speculative data.

Based on these remarks, the authors should review their thoughts and writing before resubmitting the ms.

Author Response

We thank the reviewers for the time, thoughts, and effort they put into reviewing our manuscript and for their suggestions on how we can improve it. We have revised the text on the basis of these suggestions. In addition to addressing the reviewer’s suggestions, we have added longer paragraphs on the limitations of hiPSC-based SZ modelling and a short technical section to explain the principals of hiPSC-derived oligodendrocyte precursor cell/oligodendrocyte differentiation. Moreover, the revised version has been edited. Please consider that reference numbers have changed because we have added a total of 12 references.

Response to Reviewer 2

There are many hypothetic ideas about using hiPSCs.

The isdea of the survey is great, but if the views about interractions between cells in the brain regions discussed are interesting, especially with reafgrds to neurotransmitters and other interreralationships, the imagery is poor to make such hypothesis about hiPSCs review.

We thank Reviewer 2 for highlighting the important technical and conceptual limitations of hiPSC-based SZ modeling, which we have now addressed (see response to Reviewer 1).

Readxing and reviewing some references cited, such as 24 and 25, micrographs there are deceiving because the preservation of tissue oligodendrocytes was poor and accepted for publication! The images seemed to indicate that the oligodendrocytes of normal brain patients are in fact quite similar to those of schizophrenic patients!  Moreover, would those published micrographs really represented oligodendrocytes?

We thank the reviewer for bringing to our attention the poor preservation of tissue in the electron microscopy studies by the Uranva group. Indeed, these studies were the only ones to use human electron microscopy in psychiatric disorders. We have added the following: “… because of limitations of human postmortem studies, such as longer postmortem intervals and differences in the duration of agony before death, the quality and preservation of tissue and cells in electron microscopy studies is worse in humans than in animal models. Therefore, human electron microscopy studies of the brain should be interpreted with caution.” (page 3, lines 106-109)

Additionally, to claim apoptosis...which is programmed cell death...how to justify this term and functions, especially if demyelination showed is not centrally located and without nucleus with apoptotic views? More, what would be apoptosis there, why not necrosis? how about necroptosis?

We have added the following: “To date, the presence of apoptosis or necrosis or a combination of both in the brain in severe psychiatric diseases is not fully understood. With respect to a deficit in oligodendrocytes, future studies should investigate whether cells undergo apoptosis or necrosis or whether a neurodevelopmental deficit leads to impaired differentiation, contributing to lower numbers of mature oligodendrocytes.” (page 3, lines 102-106)

The cells illustrated are not typical, certainly not for normal patients because oligodendrocytes have electron dense cytoplasms and both are clear and damaged (i.e. control vs schizophrenic ones).  Infact if the imagery and statistics can satisfy the peripheral or nerve myelinated micrographs, the support for cell bodies is none.

Please see our response to the second comment above.

Similarly, one of the reference, i.e. 32, showed in morphometry histograms, that the oligodendrocytes not dissimilar to control ones due to scattered bar of errors in measurements.

Large standard deviations are often found in structural and molecular investigations of SZ. This may be because SZ is a syndrome rather than a single disease entity and because different biological mechanisms may lead to the same symptom spectrum. Therefore, an important goal in psychiatry is to identify subgroups of patients that share biological mechanisms and symptoms (e.g. cognitive deficits). However, the samples in postmortem studies are small because of decreasing autopsy rates and the enormous effort required to collect human brains. Furthermore, because of the small sample sizes results need to be replicated in independent samples. Currently, our group (ref. 32) is performing a stereological study in the hippocampus of SZ patients vs controls in an independent sample.

I would suggest a Table 1, basec on imagery like the entire manuscript is questionable and should await further not so speculative data.

Unfortunately, we do not understand this comment. The manuscript already contains a Table 1, which summarizes the evidence on oligodendrocytes deficits in SZ from the literature.

Based on these remarks, the authors should review their thoughts and writing before resubmitting the ms.

We thank Reviewer 2 for reviewing the limitations of hiPSC-based SZ modelling and hope that the reviewer is happy with our changes and additions (see response to Reviewer 1).

Reviewer 3 Report

Line 35- ... cells (hiPSCs) to will enable functional...

Line 138 and 139- The meaning of this sentence is unclear to this reviewer. Is a particular functional deficit being alluded to? Do the authors mean that no deficit in oligodendrocyte number or myelin structure is observed? That sentence needs to be rewritten to make it meaningful. It seems to be intended as a segue to the next section but does not make sense.

Lines 174-178- The authors seem to be providing background to support  the idea that labeling NG2 cells, mature oligodendrocytes, and the stages in between should be performed in tissue from SZ patients in order to reveal the ways oligodendrocyte maturation are altred in SZ but they do  not make this point clearly or explicitly enough.

Line 228 should include a citation.

Discussion of some roles of oligodendrocyte modification of neuronal firing seem to be missing:

Battefeld, Arne, Jan Klooster, and Maarten HP Kole. "Myelinating satellite oligodendrocytes are integrated in a glial syncytium constraining neuronal high-frequency activity." Nature communications 7 (2016): 11298.

Yamazaki, Yoshihiko, et al. "Modulatory Effects of Perineuronal Oligodendrocytes on Neuronal Activity in the Rat Hippocampus." Neurochemical research 43.1 (2018): 27-40.

Author Response

We thank the reviewers for the time, thoughts, and effort they put into reviewing our manuscript and for their suggestions on how we can improve it. We have revised the text on the basis of these suggestions. In addition to addressing the reviewer’s suggestions, we have added longer paragraphs on the limitations of hiPSC-based SZ modelling and a short technical section to explain the principals of hiPSC-derived oligodendrocyte precursor cell/oligodendrocyte differentiation. Moreover, the revised version has been edited. Please consider that reference numbers have changed because we have added a total of 12 references.

Response to Reviewer 3

 Line 35- ... cells (hiPSCs) to will enable functional...

 We thank Reviewer 3 for noticing this mistake, which we have corrected.

Line 138 and 139- The meaning of this sentence is unclear to this reviewer. Is a particular functional deficit being alluded to? Do the authors mean that no deficit in oligodendrocyte number or myelin structure is observed? That sentence needs to be rewritten to make it meaningful. It seems to be intended as a segue to the next section but does not make sense.

We thank the reviewer for bringing this sentence to our attention, and we have deleted the sentence.

Lines 174-178- The authors seem to be providing background to support  the idea that labeling NG2 cells, mature oligodendrocytes, and the stages in between should be performed in tissue from SZ patients in order to reveal the ways oligodendrocyte maturation are altred in SZ but they do  not make this point clearly or explicitly enough.

We thank the reviewer for recommend more clarity. We have revised the paragraph and added the following: “It has been hypothesized that oligodendrocyte progenitor cells, which are capable of myelination, are reduced in brain regions of SZ patients, resulting in decreased plasticity and remyelination capacity. Progenitor cells can be labeled by using antibodies that bind to oligodendrocyte proteins, which are expressed during specific stages of oligodendrocyte development [43]. However, a first cell density study of the prefrontal cortex in SZ detected no loss of early NG2-immunopositive oligodendrocyte progenitor cells [46]. This study did detect a loss of oligodendrocytes positive for Olig2, a transcription factor expressed in oligodendrocyte progenitors at later stages and in mature oligodendrocytes [46], but Olig2 is not suitable for identifying progenitor cells specifically. Additional labeling with neurite outgrowth inhibitor (Nogo)-A, which reliably identifies mature oligodendrocytes, has been shown to be a way to identify specific stages of oligodendrocytes in human brain regions from patients with multiple sclerosis [47]. Nogo is known to regulate cellular processes and has three isoforms, Nogo-A, -B, and -C. Specifically, Nogo-A is highly expressed in oligodendrocytes.” (page 4, lines 176-187)

Line 228 should include a citation.

We thank the reviewer for noticing this oversight and have added the following citations:

Battefeld, A.; Klooster, J.; Kole, M.H. Myelinating satellite oligodendrocytes are integrated in a glial syncytium constraining neuronal high-frequency activity. Nat Commun 2016, 7, 11298, doi:10.1038/ncomms11298. Snaidero, N.; Simons, M. The logistics of myelin biogenesis in the central nervous system. Glia 2017, 65, 1021-1031, doi:10.1002/glia.23116.

Discussion of some roles of oligodendrocyte modification of neuronal firing seem to be missing:

Battefeld, Arne, Jan Klooster, and Maarten HP Kole. "Myelinating satellite oligodendrocytes are integrated in a glial syncytium constraining neuronal high-frequency activity." Nature communications 7 (2016): 11298.

Yamazaki, Yoshihiko, et al. "Modulatory Effects of Perineuronal Oligodendrocytes on Neuronal Activity in the Rat Hippocampus." Neurochemical research 43.1 (2018): 27-40.

We thank the reviewer for drawing our attention to these publications. We have included the suggested aspect in the section “interactions of oligodendrocytes….”. Page 6, lines 245-248: “Besides myelination and metabolic support, electrically coupled perisomatic oligodendrocytes buffer K+ currents and influence high-frequency neuronal excitability, e.g. of excitatory pyramidal [71] and hippocampal inhibitory interneurons [77].”

Round 2

Reviewer 2 Report

The lines that concerns both references 24 and 25 has been added with a fine explanation but the references are still taken in the text and Table as correct even though the ultratsructure in both published studies demonstrate no differences in human necropsies taken from normal vs schizophrenic brains.

The authors have to look at the published references and tables by 24 and 25 to convince themselves that these publications have NO values to be used as references without a more critical judgement than the mild paragraph as written that based a large part of their discussion.

Please amend.

Author Response

We thank the reviewer for their work and input. Update of the “Reference in press”: Papiol et al. 2019 is now published in Translational Psychiatry [92]. Du to reference updates, reference numbers changed.

We thank reviewer 2 for the intense review in order to improve the review. To provide an overview of the discussion we included our responses to the first review of Reviewer 2 in our review discussion. We interpretate that the comments of Reviewer 2 to our re-vision1 are referred to the references 24 and 25 and that the other issues are convincingly addressed in our revision 1.

Reviewer 2

The lines that concerns both references 24 and 25 has been added with a fine explanation but the references are still taken in the text and Table as correct even though the ultratsructure in both published studies demonstrate no differences in human necropsies taken from normal vs schizophrenic brains.

The authors have to look at the published references and tables by 24 and 25 to convince themselves that these publications have NO values to be used as references without a more critical judgement than the mild paragraph as written that based a large part of their discussion.

Answer: we would like to thank the reviewer for thoroughly and critically reading the papers. We inspected the micrographs and tables in ref. 24 and 25 and found indeed, that the differences between schizophrenia patients and normal controls are not clearly shown. Therefore we deleted these two references from the main text and the table 1. We deleted also the explanation of quality of electron microscopy in human brain tissue.

We deleted: “Moreover, electron microscopy studies have shown damaged myelin sheaths, myelin degeneration, and apoptosis/necrosis of oligodendrocytes in gray and white matter of the prefrontal cortex in SZ patients [24,25]. To date, the presence of apoptosis or necrosis or a combination of both in the brain in severe psychiatric diseases is not fully understood. With respect to a deficit in oligodendrocytes, future studies should investigate whether cells undergo apoptosis or necrosis or whether a neurodevelopmental deficit leads to impaired differentiation, contributing to lower numbers of mature oligodendrocytes. However, because of limitations of human postmortem studies, such as longer postmortem intervals and differences in the duration of agony before death, the quality and preservation of tissue and cells in electron microscopy studies is worse in humans than in animal models. Therefore, human electron microscopy studies of the brain should be interpreted with caution.”

Please amend.
